# Air Pollution Predicts Harsh Moral Judgment

**DOI:** 10.3390/ijerph16132276

**Published:** 2019-06-27

**Authors:** Hongxia Li, Xue Wang, Yafei Guo, Zhansheng Chen, Fei Teng

**Affiliations:** 1School of Economics and Management, Tsinghua University, Beijing 100084, China; 2School of Psychology, Beijing Normal University, Beijing 100875, China; 3Department of Marketing, Chinese University of Hong Kong, Hong Kong, China; 4Department of Psychology, The University of Hong Kong, Hong Kong, China; 5Center for Studies of Psychological Application, Guangdong Key Laboratory of Mental Health and Cognitive Science, The Base of Psychological Services and Counseling for “Happiness” in Guangzhou, School of Psychology, South China Normal University, Guangzhou 510631, China

**Keywords:** air pollution, morality, moral judgment, moral behavioral intention

## Abstract

The present research recruited participants from China, which is suffering from serious air pollution, and examined whether air pollution would be associated with moral judgment and immoral behavioral intention. Study 1 (*n* = 145) used the objective Air Quality Index to indicate the level of air pollution and found that it predicted harsh judgment on others’ moral violations but did not predict judgment on others’ non-moral negative behaviors or their own immoral behavioral intentions. Study 2 (*n* = 90) asked participants either to recall a past experience of being exposed to air pollution or to recall a neutral experience and consistently found that air pollution only influenced judgment on moral violations. The findings also ruled out the feeling of threat or the trust of government as possible mediators in the relationship between air pollution and harsh moral judgment.

## 1. Introduction

Abundant literature has documented the detrimental psychological outcomes of air pollution, such as emotional consequences and aggressive behaviors [1,2,3]. The current research aimed to stretch its effect to an important area in psychology—morality—and examine how air pollution would predict moral judgment and immoral behaviors.

Although the association between air pollution and moral judgment (to the scope of our knowledge) has not been examined, the symbolic relationship between physical dirtiness and moral judgment indicate their possible association. The threat to moral purity increases individuals’ desires for physical cleanliness. For example, participants who hand-copied a story of an unethical deed desired cleansing products (e.g., soap and toothpaste) more than those in the control condition [4]. Meanwhile, physical dirtiness and disgust render people to seek moral purity by upholding high moral standards and endorsing harsh moral judgment. After being exposed to a dirty desk or a disgusting odor, participants increased their unfavorable attitudes towards immoral issues, such as the marriage between first cousins or keeping a lost wallet [5].

The association between physical dirtiness and harsh moral judgment is further supported by the findings that priming participants with the concepts of physical cleanliness would soften their moral condemnation [6]. Participants in the cleanliness condition were asked to either unscramble cleanliness-related sentences or to wash their hands after watching a disgust-eliciting clip, and then they evaluated six moral dilemmas. They turned to accept those dilemmas more than those in the control condition [6]. Building on the revealed association between physical dirtiness and harsh moral judgment and the finding that air pollution increases physical dirtiness [7], we hypothesized that the severe air pollution would predict harsh moral judgment.

We were also interested in whether the harsh moral judgment associated with air pollution would spill over to judgment on non-moral negative behaviors, since the previous research had found contradictory evidence. On the one hand, Schnall and colleagues [5] instructed participants to complete tasks in a dirty room and found that the environmental dirtiness only increased condemnation on moral violations but not on non-moral issues. On the other hand, Wheatley and Haidt [8] presented hypnotized participants with a non-moral vignette in which there was a hypnotic disgust word and found that participants condemned this vignette morally wrong. Thus, the current research examines how air pollution would predict judgments on non-moral negative behaviors.

In addition, we also tested whether air pollution would predict immoral behavioral intentions. Past research found that people’s judgement of others can be inconsistent with their own behaviors. For example, the well-documented moral hypocrisy phenomenon demonstrates that people may demand others to follow moral rules or harshly condemn others’ moral violations, but they do not behave accordingly [9,10]. Therefore, the proposed prediction of air pollution on moral judgment may not be observed in the judgment of their own immoral behavioral intentions. Besides moral judgment, the current research also measured the intention of immoral behaviors. Past research has shown that physical cleanliness increased moral behaviors. Specifically, after smelling clean scents, people showed more benign behaviors, such as reciprocating others and involving in volunteering and donation, than those who did not smell clean scents [11]. Nevertheless, no evidence hitherto has linked physical dirtiness to immoral behaviors. Relevant evidence is that air pollution positively relates to aggressive behaviors, such as high rate of family disturbances [12]. However, immoral behaviors—as an umbrella concept—consist of both aggressive (e.g., hurting others) and non-aggressive behaviors (e.g., stealing). It is still unclear whether air pollution would link to all kinds of immoral behaviors, thus we examine their relationship in the current research.

To explore how air pollution influences people’s judgment on others’ moral violations, others’ non-moral negative behaviors, and their own immoral behavioral intentions, Study 1 recruited participants from Beijing during and right after the Asia-Pacific Economic Cooperation (APEC) summit 2014 and used the objective Air Quality Index (AQI) to indicate the level of air pollution. We planned to collect data during this period because the summit took place during the autumn haze episode of Beijing. Due to the Chinese government’s emission reduction campaign, Beijing did witness a rare blue sky during the summit. The good weather, however, did not last long, and the air quality fluctuated greatly in a short time, which provided a rare opportunity to test the association between objective air pollution and morality. In Study 2, we manipulated participants’ exposure to air pollution and directly tested the causal relationship between air pollution and morality. These two studies used distinct measures to assess participants’ judgments on others’ moral violations, others’ non-moral negative behaviors (to test the spillover effect), and participants’ own immoral behavioral intentions. The results consistently found that severe air pollution predicted harsh judgment on others’ moral violations but did not predict judgment on others’ non-moral negative behaviors or their own immoral behavioral intentions.

We predetermined the required sample size for two studies by G*Power 3.1 [13]. Because there were no prior studies examining the association between air pollution and moral judgment, we estimated the effect size of Study 1 to be small (*η*^2^ = 0.05). Accompanying an α of 0.05 (two tailed) and a power of 0.80, the required sample size of Study 1 was 159. In Study 2, due to the manipulation of air pollution, we predicted its effect size to be increased (*η*^2^ = 0.10). With an α of 0.05 (two tailed) and a power of 0.80, the required sample size of Study 2 was 81. In addition, the data collection was further contingent on participants’ availability. We finally recruited 145 participants for Study 1 and 90 participants for Study 2.


**Study 1**


## 2. Method

### 2.1. Participants and Design

One hundred and forty five Chinese students (124 women; age 20–42, *M*_age_ = 22.54 years, *SD*_age_ = 3.50) from a university in Beijing, China were recruited to participate in the research for partial course credits. As mentioned above, participants were required to complete an online survey during and right after APEC when they were in Beijing (please refer to Appendix A for the detailed data collection). The survey was administered by Qualtrics.

After finishing data collection of the online survey, we downloaded the hourly AQI of Beijing from the website of Ministry of Environment Protection of China (http://datacenter.mep.gov.cn/report/air_daily/airDairyCityHourMain.jsp?lang=) and matched the hourly AQI to each participant according to the time they completed the survey. The downloaded AQI is calculated based on the concentrations of six major air pollutants (i.e., CO, SO_2_, PM10, O_3_, NO_2_, and PM_2.5_ [14]), and higher AQI indicates more severe air pollution.

### 2.2. Procedure and Materials of the Online Survey

After completing the informed consent, participants first answered the question, “What do you think about the weather today?” on a two-item scale (1 = *very bad/unpleasant*, 7 = *very good/pleasant*). The scores were averaged to indicate the subjective evaluation of weather (*M* = 4.87, *SD* = 1.62, *r* = 0.90). It is noteworthy that the measure of weather is different from air pollution index because many factors other than air pollution (e.g., cloud, rain, and wind) can influence weather evaluation. We intended to measure and control for the general weather to observe the exclusive effect of air pollution. To support our assumption that the subjective evaluation of weather and air pollution is different, Table 1 shows that they only had a small and weak correlation (*r* = 0.10).

Participants then reported their current mood on six items (*Currently I feel happy/cheerful/satisfied/sad/unhappy/dejected*; 1 = *not at all*, 7 = *very much* [10]). Scores of positive and negative mood were averaged separately, with higher scores indicating more salient mood (positive mood: *M* = 4.34, *SD* = 1.58, *α* = 0.96; negative mood: *M* = 2.36, *SD* = 1.33, *α* = 0.88). Participants also evaluated how *threatened*, *uncontrollable,* and *unsafe* (1 = *not at all*, 7 = *very much*) they felt at that moment. The scores were averaged to index the feeling of threat (*M* = 2.01, *SD* = 1.25, *α* = 0.90).

Next, participants proceeded to the measure of moral judgment and indicated how they accepted five vignettes in which an anonymous person commits immoral behaviors on a 9-point scale (1 = *definitely unacceptable*, 9 = *definitely acceptable*). The first four vignettes were immoral behaviors adopted from previous research [10], including breaking traffic rules for a late appointment, keeping a stolen and abandoned bike, picking up a wallet and keeping money inside, and plagiarizing in a test. The scores of these four vignettes were averaged to index judgment on other people’s immoral behaviors (*M* = 2.89, *SD* = 1.43, *α* = 0.64). The low Cronbach’s alpha may be because these moral vignettes gauged different aspects of morality. To test the spillover effect, participants also read an anonymous person’s negative behavior that did not violate moral rules (please refer to Appendix B for the vignette) [5] and indicated their degree of acceptance on the same 9-point scale (*M* = 6.65, *SD* = 2.26).

In addition, a hypothetical negotiation task was used to measure participants’ dishonest behavioral intentions [15,16]. Participants imagined that they were human resource managers and needed to negotiate a salary with a job candidate. If they lied to the candidate, they might receive an end-of-year bonus. Otherwise, their annual performance might be evaluated lower. Then, participants were asked to indicate the percentage of chance (from 0 to 100) that they would not tell the lie (*M* = 35.06, *SD* = 28.83), with lower scores indicating higher dishonest behavioral intention.

As social class could influence moral judgment [15], we asked participants to report their subjective social class on the MacArthur Scale of Subjective Socioeconomic Status (SES) [17]. Specifically, they selected a class on a 10-rung ladder based on their standing in the society relative to others (1 = *lowest standing*, 10 = *highest standing*; *median* = 5.00, *mean* = 5.37, *SD* = 1.49). A debriefing followed.

## 3. Results

Hourly AQI did not significantly predict positive mood, negative mood, or the feeling of threat (*p*s > 0.05; see Table 1 for the correlation matrix).

### 3.1. Moral Judgment

The full regression results are presented in Table 2. The averaged acceptance of four moral violations was regressed onto hourly AQI while controlling for positive mood, negative mood, subjective evaluation of the weather that day, social class, and the dummy-coded gender (male = 1, female = 0). The model that only included the control variables was significant (*F*(5, 139) = 3.59, *p* < 0.01) and explained 11.4% of the variance. After adding hourly AQI as a predictor, the whole model was still significant (*F*(6, 138) = 3.78, *p* < 0.01) and explained 14.1% of the variance. Consistent with our hypothesis, higher hourly AQI predicted lower acceptance of moral violations (*b* = −0.01, *SE* = 0.003, *t* = −2.07, *p* < 0.05, *f*^2^ = 0.03). Among all the control variables, only negative mood (*b* = 0.27, *SE* = 0.09, *t* = 2.92, *p* < 0.01) and the subjective evaluation of weather that day (*b* = 0.25, *SE* = 0.08, *t* = 3.13, *p* < 0.01) were significant predictors, such that participants with more negative mood or those evaluating the weather better accepted the moral violations more.

Then, the same procedure was processed to test whether hourly AQI predicted judgment on negative behaviors that did not violate moral rules (i.e., laziness) while controlling for the same variables. The model that only included the control variables was significant (*F*(5, 139) = 5.45, *p* < 0.001) and explained 16.4% of the variance. After adding hourly AQI as a predictor, the whole model was significant (*F*(6, 138) = 4.65, *p* < 0.001) and explained 16.8% of the variance. However, hourly AQI did not significantly predict the acceptance (*p* > 0.05). More negative mood (*b* = −0.29, *SE* = 0.15, *t* = −1.99, *p* < 0.05) and the better subjective evaluation of weather of that day (*b* = 0.40, *SE* = 0.13, *t* = 3.20, *p* < 0.01) predicted less acceptance of laziness. We also found a gender difference that females accepted laziness more than males (*b* = −1.48, *SE* = 0.51, *t* = −2.90, *p* < 0.01). Therefore, participants immersed in polluted air increased their condemnation on immoral behaviors but not on non-moral negative behaviors.

### 3.2. Telling the Truth

The percentage of chance to tell the truth was regressed onto hourly AQI while controlling for the same confounding variables as in the above analyses. The model that only included the control variables was not significant (*F*(5, 139) = 0.72, *p* > 0.05) and explained 2.6% of the variance. After adding hourly AQI as a predictor, the whole model was still not significant (*F*(6, 138) = 0.88, *p* > 0.05) and explained 3.7% of the variance. Neither hourly AQI nor any control variables were significant predictors of the dishonest intentions (*p*s > 0.05).

### 3.3. Mediation of the Feeling of Threat

The above findings revealed that air pollution only significantly predicted harsh judgment on immoral acts but not judgments on non-moral behaviors or dishonest intentions. We further tested whether the feeling of threat would be a possible mediator of the relationship between air pollution and moral judgment.

By regressing the feeling of threat on hourly AQI while controlling for other control variables, we found that the whole model was significant (*F*(6, 138) = 19.05, *p* < 0.001) and accounted for 45.3% variance. More negative mood (*b* = 0.56, *SE* = 0.07, *t* = 8.60, *p* < 0.001) and worse subjective evaluation of weather that day (*b* = −0.16, *SE* = 0.06, *t* = −2.81, *p* < 0.01) were associated with higher threatened feelings (for other control variables, *p*s > 0.05). However, hourly AQI was not a significant predictor (*p* > 0.05). Via bootstrap analysis [18], we did not find a significant indirect effect of the feeling of threat on the association between hourly AQI and moral judgment, because the 95% bias-corrected confidence interval included zero (95% CI: (−0.00, 0.00)).

Study 1 provided correlational evidence that high air pollution predicted harsh judgment on immoral acts but not judgments on non-moral negative behavior or immoral behavioral intention. It also ruled out the feeling of threat as a possible mediator to account for the association between air pollution and harsh moral judgment. Nevertheless, Study 1 had several limitations warranting attention. First, the results were only correlational evidence. Second, it measured judgments on non-moral negative behavior and immoral behavioral intention by only one item each, limiting the generalization of the findings.

As a necessary extension, Study 2 manipulated participants’ exposure to air pollution. Furthermore, participants’ trust of their government was measured, because the government should take responsibility for the bad air quality in many cases. Taking China as an example, among all the sources of air pollution in China, anthropogenic factors such as power plants and industrial boilers contribute the most [19]. Exposure to air pollution might reduce trust in government, which further decreases the intolerance of moral violations in general.


**Study 2**


## 4. Method

### 4.1. Participants and Design

Ninety participants (61 women; age 14–46, *M*_age_ = 26.97 years, *SD*_age_ = 7.09) were recruited from a university in Beijing, China for partial course credits. Participants were randomly assigned to either air pollution condition or control condition. They completed the tasks during the class and on computers. All the tasks were administered by Qualtrics.

### 4.2. Procedure and Materials

After completing an online informed consent form, participants first received the manipulation of exposure to air pollution. Participants in the air pollution condition were asked to recall a recent experience on a hazy and polluted day. Then, they used at least 50 words to describe the weather and how they felt on that day. Participants in the control condition were asked to choose one day in the past week and describe what they had done on that day with at least 50 words. We checked participants’ answers and found that all the participants followed the instructions.

Participants then reported their current moods. In addition to the six items used in Study 1, two additional items were added (i.e., *angry* and *anxious*; 1 = *not at all*, 7 = *very much*). Scores on three positive-mood items (*M* = 4.05, *SD* = 1.47, *α* = 0.89) and five negative-mood items (*M* = 2.93, *SD* = 1.56, *α* = 0.93) were averaged.

Six moral violations were adapted from previous studies to measure moral judgment [20], including, “In order to attend an important interview on time, one doesn’t help a stranger fainted on the roadside”, “In order to graduate, one plagiarizes graduation thesis from others’ papers”, “One forbids his wife/her husband to wear clothing that he/she has not first approved”, “One tries to undermine all of his/her boss’ ideas in front of others”, “One always said publicly that his/her own home country is worse than other countries”, and “One gets married to his/her cousin.” For the two vignettes of non-moral negative behaviors, we retained the vignette of laziness from Study 1 and created another vignette, “One paints his rooms’ walls blue although all of his friends disapprove of it”.

Participants reported how acceptable they thought these behaviors were (1 = *definitely unacceptable*, 7 = *definitely acceptable*). The acceptance of six moral violations (*M* = 3.20, *SD* = 0.93, *α* = 0.41) and those of two non-moral negative behaviors (*M* = 5.66, *SD* = 1.39, *r* = 0.37) were averaged (please refer to Appendix C for the possible explaination of the low internal consistency or correlation).

Two tasks were used to measure immoral behavioral intentions. In the first task, participants read six hypothetically unethical behaviors adopted from previous research [15], including “Use office supplies, Xerox machine, and stamps for personal purposes”, “Make personal long-distance phone calls at work”, “Waste company time surfing on the internet, playing computer games, and socializing”, “Take merchandise and/or cash home”, “Abuse the company expense accounts and falsify accounting records”, and “Receive gifts, money, and loans (bribery) from others due to one’s position and power”. Participants reported the possibility that they would engage in these behaviors (1 = *not likely at all*, 7 = *very likely*). The scores were averaged to indicate their immoral behavioral intentions (*M* = 2.61, *SD* = 1.16, *α* = 0.82).

The second task was to allocate two tasks between oneself and other people [21]. Participants were told that they would complete a five-minute task at the end of the research, and there were two alternative tasks. One task was easier and more interesting than the other. Participants were told that the program would randomly choose some decision-makers. If chosen as a decision-maker, the participant could distribute one task to oneself and leave the other to one of the following participants; otherwise, they would complete a task left by a previous decision-maker. Participants were also told that their roles and choices were anonymous. Actually, all the participants were designated as decision-makers. If assigning the easier task to oneself, participants’ behavior would be categorized as immoral and selfish.

After completing the above measures, participants evaluated how dirty the weather was in their recalled scenarios (1 = *very dirty*, 7 = *very clean*), with lower scores indexing more perceived dirtiness (*M* = 3.92, *SD* = 2.04). The question intended to check the effectiveness of air pollution manipulation.

As in Study 1, we asked participants to evaluate the current weather (1 = *very bad*, 7 = *very good*; *M* = 4.68, *SD* = 1.46). Given that washing behaviors might influence their moral judgment and moral behaviors [6,11], they also reported whether they had washed themselves (e.g., washing hands, having a bath, etc.) right before attending the research by choosing between *yes* or *no*.

Then, we measured participants’ trust in government with the following statements: “I feel that government acts in citizen’s best interest”, “I feel fine interacting with the government since the government generally fulfills its duties efficiently”, “I always feel confident that I can rely on government to do their part when I interact with them”, and “I am comfortable relying on the government to meet their obligations” [22]. Participants reported their agreements on the above items (1 = *completely disagree*, 7 = *completely agree*). Scores were averaged, with higher scores indicating more trust in government (*M* = 4.09, *SD* = 1.57, *α* = 0.93).

We further used the 4-item Perceived Awareness of the Research Hypothesis scale (PARH; [23]) to measure whether participants had detected any hypothesis of the research. Sample items included, “I knew what the researchers were investigating in this research”, and “I wasn’t sure what the researchers were trying to demonstrate in this research” (1 = *completely disagree*, 7 = *completely agree*). Scores were reversed, if necessary, and averaged. Higher scores indicated more detection of the research hypothesis (*M* = 3.84, *SD* = 1.38, *α* = 0.81).

Finally, participants reported their genders and ages. As Study 1 did not find social class as a significant confounding variable, we did not include it in Study 2. Participants received debriefing after completing the whole research.

## 5. Results

Please see Table 3 for the correlation matrix. Independent *t*-tests showed that people in the air pollution condition reported less positive mood (*M* = 3.63, *SD* = 1.38) than those in the control condition (*M* = 4.39, *SD* = 1.46, *t* = 2.49, *p* < 0.05, Cohen’s *d* = 0.53), but two groups did not show any significant differences in their negative mood, trust in government, or scores of PARH (*p*s > 0.05).

We also found that participants in the air pollution condition perceived the air as less clean and more dirty (*M* = 2.58, *SD* = 1.89) than those in the control condition (*M* = 5.00, *SD* = 1.43, *t* = 6.72, *p* < 0.001, Cohen’s *d* = 1.45). Therefore, exposure to air pollution effectively increased participants’ perception of dirtiness.

### 5.1. Moral Judgment

The full regression results are presented in Table 4. The judgment on moral violations was regressed onto the dummy code of manipulation conditions (air pollution = 1, control = 0) while controlling for positive mood, negative mood, current weather, the dummy code of gender (male = 1, female = 0), and the dummy code of washing oneself before the research (yes = 1, no = 0). It revealed that participants in the air pollution condition accepted the moral violations less (*M* = 2.90, *SD* = 0.83) than those in the control condition (*M* = 3.34, *SD* = 0.83; *b* = −0.40, *SE* = 0.19, *t* = −2.14, *p* < 0.05, *f*^2^ = 0.06), supporting our hypothesis that exposure to air pollution decreased acceptance of moral violations. Meanwhile, higher negative mood (*b* = 0.14, *SE* = 0.06, *t* = 2.28, *p* < 0.05) predicted more acceptance of moral violations, which was consistent with the finding of Study 1. No other control variables were significant predictors (*p*s > 0.05).

The same regression was processed while using judgment on non-moral negative behaviors as the outcome. It revealed that better current weather predicted more acceptance (*b* = 0.20, *SE* = 0.09, *t* = 2.15, *p* < 0.05). Meanwhile, participants who had washed themselves before the research accepted the non-moral negative behaviors more than those had not washed themselves beforehand (*b* = 0.61, *SE* = 0.28, *t* = 2.18, *p* < 0.05). However, the manipulation did not significantly influence the judgment (*p* > 0.05), nor did other control variables (*p*s > 0.05).

### 5.2. Immoral Behavioral Intentions and Task Allocation

We regressed participants’ likelihood to commit eight immoral behaviors on the dummy code of manipulation conditions (air pollution = 1, control = 0) while controlling for the same control variables as in the above analyses. Neither the manipulation (*p* > 0.05) nor the control variables (*p*s > 0.05) significantly predicted the intention to act immorally.

Then, we compared the task allocations between participants in two conditions; 26 out of 40 participants (65%) in the air pollution condition and 39 out of 50 participants (78%) in the control condition distributed the easier task to themselves and left the harder one to others. Pearson’s Chi-Squared test revealed that the manipulation conditions and participants’ task allocations were independent (*χ*^2^ = 1.87, *p* > 0.05), suggesting that the exposure to air pollution did not influence how participants allocated the tasks.

### 5.3. Mediation of Trust in Government

An independent *t*-test showed that the exposure to air pollution did not significantly influence participants’ trust in the government (air pollution: *M* = 3.98, *SD* = 1.68; control: *M* = 4.19, *SD* = 1.48, *t* = 0.63, *p* > 0.05). However, lower trust in the government was correlated to higher acceptance of immoral acts (*r* = −0.28, *p* < 0.01), thus people who trusted in the government less were more likely to tolerate immoral behaviors and less likely to condemn them.

We further tested whether the trust in government would account for the influence of air pollution on moral judgment. We also controlled for all the control variables as in the above analyses. The results of bootstrap analysis [18] showed that trust in government was not a significant mediator, because the 95% bias-corrected confidence interval of its indirect effect included zero (95% CI: (−0.08, 0.18)).

Taken together, Study 2 showed that the exposure to air pollution decreased participants’ acceptance of immoral behaviors but did not influence their acceptance of non-moral negative behaviors or their own immoral behavioral intentions, which paralleled well to the findings of Study 1. Meanwhile, it ruled out the possibility that participants’ trust in government mediated the relationship between air pollution and harsh moral judgment. It also demonstrated that exposure to air pollution effectively made participants perceive high physical dirtiness.

## 6. Discussion

Two studies consistently found that air pollution predicted harsh judgment on moral violations, but it had no influence on the judgment of non-moral negative behaviors, which did not support the spillover effect [5,8]. Meanwhile, air pollution was not a significant predictor of immoral behavioral intentions, which were measured by self-reported intentions to act dishonestly, immorally, and selfishly. It is noteworthy that we used an objective AQI to index air pollution in Study 1, which increased the ecological validity of the findings. Study 2 further manipulated participants’ exposure to air pollution, revealing causal evidence that air pollution could result in increased condemnation of immoral behaviors, which dovetails nicely with previous evidence that moral condemnation was aggravated by the dirty environment [5].

Meanwhile, both studies controlled for some variables such as gender, mood, and subjective evaluation of the weather. In doing so, we excluded the possible effects of some individual bias and confounding variables and mainly focused on the influence of air pollution.

### 6.1. Theoretical Contributions and Implications

Our findings extend the current understanding of the effects of air pollution. Previous studies mainly documented its association with physical health, such as increased rate of disease and mortality [24]. Although there is also research associating air pollution with psychological factors, it predominantly focuses on the negative outcomes, such as emotional symptoms and aggression [1,2]. The current research showed that air pollution could increase the harshness of moral judgment, which not only bridges the gap between air pollution and morality but also reveals a positive outcome of air pollution—that it renders people to condemn moral violations and safeguard moral rules.

The current research also ruled out two possible explanations for our reported effect of air pollution on harsh moral judgment. Severe air pollution may lead people to feel threatened and be more alarmed to potential threats and thus decrease people’s tolerance of moral misconducts. However, the findings of Study 1 ruled out this possibility and revealed that the level of objective air pollution did not predict the feeling of threat, which did not mediate the relationship between air pollution and harsh moral judgment either. The above findings may contribute to our special sample. All of our participants in Study 1 were from Beijing and have been exposed long-term to air pollution, thus they may have become used to the frequent and long-lasting haze weather and perhaps have developed a strategy to overcome the unsafe and threatening feelings derived from air pollution.

Study 2 tested whether the manipulation of exposure to air pollution would decrease trust in government and whether it would further increase moral condemnation. We found that participants’ trust in government was not influenced by the manipulation, and the trust in government could not explain the reported effect of air pollution on moral judgment. It is noteworthy that we found lower trust in government associated with higher acceptance of immoral behaviors.

The current research provides additional evidence for the parallel relationship between physical and psychological purity. Although a body of evidence has shown that physical dirtiness could induce harsh moral judgment, most research generated dirtiness through particular senses, such as visualization, taste, or gustation [5,25]. The current research showed that air pollution could be a source of physical dirtiness, and it could threaten moral purity and increase moral condemnation.

By demonstrating that air pollution could increase perceived physical dirtiness, the current research opens up a host of possible psychological outcomes of air pollution. For example, previous research revealed that reminders of physical purity influenced political and attitudes towards homosexuality. In particular, people who were reminded of the hand-sanitizer dispenser or participated in hand-wiping reported more conservative political orientation [26]. Likewise, the personal sensitivity of feelings of disgust as well as ambient, disgusting smells was associated with the disapproval of homosexuality [27,28]. Given that air pollution makes people perceive more physical dirtiness, it might influence people’s attitudes towards conservative political orientation and homosexuality.

Our findings may also carry implications on the jury system. Many countries, such as America and England, implement a jury system in which a jury can make the decision of cases and direct the actions of the judge. Although our research only recruited participants from China, air pollution as a global problem should exert influence not only on Chinese but also on residents in other polluted countries. Based on our findings, it is plausible to assume that the exposure to air pollution will increase the harshness of moral judgment whether it be among Westerners or Easterners. Thus, jury evaluations and decisions of cases are very likely to be influenced and even biased by the outside air quality. This possibility awaits future attention.

### 6.2. Limitations and Future Directions

Nonetheless, the current research has some limitations that warrant attention. First, although we ruled out two factors, namely the feeling of threat and the trust of government, as possible mediators, it is still unclear why air pollution induces harsh moral judgment. Future research may examine other possible mediators, such as increased concern for moral rules or need for moral purity.

Second, with the assumption that people who have experienced polluted air might have a strong reaction to air pollution, we collected data for both studies from China, which has been undergoing lasting and severe air pollution. Consistent with the previous studies’ effect sizes [6] (ranging from 0.05 to 0.11), our findings had similarly small effects (Study 1, *f*^2^ = 0.03; Study 2, *f*^2^ = 0.06). One possible account for this small effect might be that Chinese citizens apply strategies to alleviate the influence of air pollution and gradually get used to or even become indifferent to it. In contrast, those who have seldom or never been exposed to air pollution may react intensely because they are more sensitive to it. This possibility is supported by the evidence that, compared to frequent players, inexperienced players felt more morally distressed when playing a violent game because they were more sensitive [29].

Lastly, both Studies 1 and 2 found that negative mood positively predicted the harsher judgment of others’ immoral behaviors, which seems inconsistent with past research. Previous research showed that some negative emotions (e.g., disgust and anger) were associated with more critical or harsher moral judgment. However, not all negative emotions are associated with harsher judgments of others. For instance, Schnall et al. [6] found that induced sadness (also measured in the present research) could result in less harsh moral judgment when compared to the neutral condition. In other words, the relationship between negative emotion and moral judgment might vary as a function of each discrete emotion and moral domain [30,31]. Morever, the past research on the relationship between negative emotion and moral judgment mostly focused on one specific emotion, such as anger or disgust. Little research (in the scope of our knowledge) has examined the impact of general negative mood on moral judgment. It might be an interesting direction for future research to test the relationship between general negative mood and moral judgment and the psychological mechanism.

## 7. Conclusions

To conclude, the current research showed that high objective Air Quality Index and exposure to air pollution were positively associated with harsh judgment on others’ moral violations but were not associated with judgment on others’ non-moral violations or their own immoral behavioral intentions. Moreover, neither the feeling of threat nor the trust in government could explain the relationship between severe air pollution and harsh moral judgment. These findings not only provide additional evidence for the symbolic association between physical and psychological purity but also extend our understandings of the psychological effects of air pollution.

## Figures and Tables

**Table 1 ijerph-16-02276-t001:** The correlations of variables used in Study 1.

Variables	Acceptance of Moral Violations	Acceptance of Non-Moral Behaviors	Telling Truth	Positive Mood	Negative Mood	Weather	Social Class	Gender ^†^
Hourly AQI	−0.11	−0.06	0.12	−0.09	0.11	0.10	−0.06	0.04
Acceptance of moral violations		0.15	0.25 **	−0.01	0.17 *	0.18 *	0.16	0.08
Acceptance of non-moral behaviors			0.05	0.12	−0.23 **	0.30 **	0.08	−0.23 **
Telling truth				−0.10	0.09	−0.10	−0.12	0.01
Positive mood					−0.34 **	0.42 **	0.26 **	−0.21 *
Negative mood						−0.31 **	−0.04	0.01
Weather							0.20 *	−0.07
Social class								−0.10

* Correlation is significant at the 0.05 level (2-tailed). ** Correlation is significant at the 0.01 level (2-tailed). ^†^ Gender was coded as male = 1, female = 0. AQI: Air Quality Index.

**Table 2 ijerph-16-02276-t002:** The full regression results of Study 1.

Dependent Variable and Predictors						
**Acceptance of Moral Violations as the Dependent Variable**						
**Variables**	**Unstandardized *b***	**Unstandardized *SE*^†^**	**Standardized *b***	***t***	***p***	**95% CI**
Intercept	1.02	0.67		1.53	0.129	(−0.3019, 2.3416)
Positive mood	−0.06	0.09	−0.07	−0.71	0.477	(−0.2284, 0.1074)
Negative mood	0.27	0.09	0.25	2.92	0.004	(0.0883, 0.4574)
Weather	0.25	0.08	0.28	3.13	0.002	(0.0924, 0.4099)
Social class	0.13	0.08	0.13	1.60	0.113	(−0.0305, 0.2846)
Gender	0.41	0.33	0.10	1.26	0.21	(−0.2356, 1.0641)
Hourly AQI	−0.01	0.00	−0.17	−2.07	0.041	(−0.0111, −0.0002)
**Acceptance of Non-Moral Negative Behavior as the Dependent Variable**						
**Varaibles**	**Unstandardized *b***	**Unstandardized *SE***	**Standardized *b***	***t***	***p***	**95% CI**
Intercept	6.52	1.04		6.28	<0.001	(4.4650, 8.5680)
Positive mood	−0.17	0.13	−0.12	−1.31	0.193	(−0.4332, 0.0881)
Negative mood	−0.29	0.15	−0.17	−1.99	0.049	(−0.5743, −0.0014)
Weather	0.40	0.13	0.29	3.20	0.002	(0.1526, 0.6453)
Social class	0.02	0.12	0.02	0.20	0.845	(−0.2203, 0.2687)
Gender	−1.48	0.51	−0.23	−2.90	0.004	(−2.4878, −0.4705)
Hourly AQI	0.00	0.00	−0.07	−0.84	0.405	(−0.0120, 0.0049)
**The Percentage of Chance to Tell the Truth as the Dependent Variable**						
**Variables**	**Unstandardized *b***	**Unstandardized *SE***	**Standardized *b***	***t***	***p***	**95% CI**
Intercept	44.24	14.25		3.11	0.002	(16.0678, 72.4123)
Positive mood	−0.44	1.81	−0.02	−0.25	0.807	(−4.0225, 3.1364)
Negative mood	0.99	1.99	0.05	0.50	0.621	(−2.9477, 4.9191)
Weather	−1.13	1.71	−0.06	−0.66	0.510	(−4.5125, 2.2540)
Social class	−1.88	1.70	−0.10	−1.11	0.270	(−5.2383, 1.4778)
Gender	−1.56	7.01	−0.02	−0.22	0.824	(−15.4133, 12.2896)
Hourly AQI	0.08	0.06	0.11	1.27	0.205	(−0.0413, 0.1908)

**^†^** SE: standard deviation.

**Table 3 ijerph-16-02276-t003:** The correlations of variables used in Study 2.

Variables	Acceptance of Moral Violations	Acceptance of Non-Moral Behaviors	Immoral Behavioral Intention	Task Allocation ^†^	Positive Mood	Negative Mood	Weather	Wash ^†^	Gender ^†^
Manipulation ^†^	−0.26 *	−0.13	−0.09	−0.14	−0.26 *	0.11	−0.15	−0.07	0.14
Acceptance of moral violations		0.24 *	0.42 **	−0.07	0.08	0.16	0.02	−0.00	−0.20
Acceptance of non-moral behaviors			0.17	−0.08	0.04	−0.01	0.19	0.16	0.15
Immoral behavioral intention				−0.11	−0.05	0.08	0.17	0.03	−0.05
Task allocation					−0.03	0.06	0.11	−0.04	−0.00
Positive mood						−0.42 **	−0.07	−0.02	0.07
Negative mood							0.08	0.07	0.07
Weather								−0.28 **	0.03
Wash ^†^									−0.09

* Correlation is significant at the 0.05 level (2-tailed). ** Correlation is significant at the 0.01 level (2-tailed). ^†^ Manipulation conditions were coded as air pollution = 1, control = 0. Task allocation was coded as easy task to oneself = 1, easy task to others = 0. Gender was coded as male = 1, female = 0. Wash was coded as yes = 1, no = 0.

**Table 4 ijerph-16-02276-t004:** The full regression results of Study 2.

Dependent Variable and Predictors						
**Acceptance of Moral Violations as the Dependent Variable**						
**Variables**	**Unstandardized *b***	**Unstandardized *SE***	**Standardized *b***	***t***	***p***	**95% CI**
Intercept	3.37	0.59		5.75	<0.001	(2.2027, 4.5348)
Positive mood	0.08	0.07	0.13	1.15	0.253	(−0.0570, 0.2133)
Negative mood	0.14	0.06	0.26	2.28	0.025	(0.0183, 0.2651)
Weather	−0.02	0.06	−0.04	−0.38	0.704	(−0.1397, 0.0948)
Gender	−0.37	0.19	−0.20	−1.95	0.055	(−0.7465, 0.0078)
Wash	−0.11	0.18	−0.06	−0.58	0.566	(−0.4693, 0.2584)
Manipulation	−0.40	0.19	−0.23	−2.14	0.036	(−0.7709, −0.0277)
**Acceptance of Non-Moral Negative Behaviors as the Dependent Variable**						
**Variables**	**Unstandardized *b***	**Unstandardized *SE***	**Standardized *b***	***t***	***p***	**95% CI**
Intercept	3.93	0.90		4.37	<0.001	(2.1420, 5.7234)
Positive mood	0.01	0.10	0.01	0.09	0.933	(−0.1987, 0.2164)
Negative mood	−0.03	0.10	−0.04	−0.35	0.725	(−0.2231, 0.1559)
Weather	0.20	0.09	0.24	2.15	0.035	(0.0144, 0.3747)
Gender	0.48	0.29	0.17	1.64	0.105	(−0.1015, 1.0569)
Wash	0.61	0.28	0.24	2.18	0.032	(0.0545, 1.1721)
Manipulation	−0.25	0.29	−0.10	−0.86	0.390	(−0.8184, 0.3229)
**Immoral Behavioral Intentions as the Dependent Variable**						
**Variables**	**Unstandardized *b***	**Unstandardized *SE***	**Standardized *b***	***t***	***p***	**95% CI**
Intercept	1.60	0.84		1.90	0.060	(−0.0717, 3.2636)
Positive mood	0.08	0.10	0.10	0.84	0.405	(−0.1120, 0.2746)
Negative mood	0.08	0.09	0.11	0.91	0.368	(−0.0962, 0.2567)
Weather	0.14	0.08	0.19	1.62	0.110	(−0.0315, 0.3040)
Gender	−0.14	0.27	−0.06	−0.52	0.607	(−0.6793, 0.3994)
Wash	0.15	0.26	0.07	0.59	0.560	(−0.3673, 0.6735)
Manipulation	−0.09	0.27	−0.04	−0.32	0.746	(−0.6181, 0.4448)

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
