# Peer review of "Air Pollution Predicts Harsh Moral Judgment"

_ijerph, 2019, doi:10.3390/ijerph16132276_

Round 1
Reviewer 1 Report
The study examined the effect of air pollution on moral judgment. It was found that in the presence of other factors, air pollution could predict moral judgment but not non-moral negative behavior or moral intention. Overall, I found the study interesting. I appreciate that the authors used real air pollution index to predict moral judgment. That said, there are some issues I hope the authors can address to improve the manuscript.
1. Consistently, the authors found that air pollution (real index in Study 1 and imagined in Study 2) could predict immoral judgment but not immoral intention. While the finding was interesting, I was not sure about the rationale behind the finding. What was the fundamental difference between the two variables? In both studies, immoral judgment was made toward other people, whereas immoral intention was measured based on participants’ motivation. Hence, was the difference between the two variables due to self vs. other? I think in Introduction you should say something about the differences between the two dependent variables.
2. While I believe it was correct to use regression, presenting the correlational matrix also helps. For example, it is of interest to know whether moral judgment and moral intention correlate or not.
3. Study 2. How were participants recruited? Air pollution varied a lot in China. Were participants recruited from difference regions? Additionally, did all participants complete the survey at the same time? If not, participants might experience difference levels of air pollution when they were completing the survey. These two factors might confound your findings. As an example, the variable of current weather was not significant in Study 1. However, the current weather, as indicated by AQI, was significant in Study 1. Thus, the findings appeared inconsistent between the two studies.
4. Even you assigned someone to the control condition, what if someone recalled a day with air pollution?
5. Abstract. You stated that “Study 1 used the object Air Quality Index to indicate the level of air pollution.” While your statement was correct, you should also indicate the main findings of study 1.
Author Response
Reviewer1:Comments and Suggestions for Authors
The study examined the effect of air pollution on moral judgment. It was found that in the presence of other factors, air pollution could predict moral judgment but not non-moral negative behavior or moral intention. Overall, I found the study interesting. I appreciate that the authors used real air pollution index to predict moral judgment. That said, there are some issues I hope the authors can address to improve the manuscript.
Question 1: Consistently, the authors found that air pollution (real index in Study 1 and imagined in Study 2) could predict immoral judgment but not immoral intention. While the finding was interesting, I was not sure about the rationale behind the finding. What was the fundamental difference between the two variables? In both studies, immoral judgment was made toward other people, whereas immoral intention was measured based on participants’ motivation. Hence, was the difference between the two variables due to self vs. other? I think in Introduction you should say something about the differences between the two dependent variables.
Answer question: Thank you for your valuable suggestion. We agreed with you that moral judgment and immoral intention are fundamentally distinct due to the self vs. other difference. Accordingly, we have added a detailed explanation of this fundamental difference in the Introduction. The specific content is as follows: “In addition, we also tested whether air pollution would predict immoral behavioral intentions. Past research found that people’s judgement on others can be inconsistent with their own behaviors. For example, the well-documented moral hypocrisy phenomenon demonstrates that people may demand others to follow moral rules or harshly condemn others’ moral violations, but they don’t behave accordingly (i.e., moral hypocrisy; e.g., Batson, Thompson, & Chen, 2002; Lammers, Stapel, & Galinsky, 2010). Therefore, the proposed prediction of air pollution on moral judgment may not be observed in the judgment of their own immoral behavioral intentions. Besides moral judgment, the current research also measured the intention of immoral behaviors. Past research has shown that physical cleanliness increased moral behaviors. Specifically, after smelling clean scents, people showed more benign behaviors, such as reciprocating others and involving in volunteering and donation, than those who did not smell clean scents (Liljenquist, Zhong, & Galinsky, 2010). Nevertheless, no evidence hitherto has linked physical dirtiness to immoral behaviors. Relevant evidence is that air pollution positively relates to aggressive behaviors, such as high rate of family disturbances (Rotton & Frey, 1985). However, immoral behaviors, as an umbrella concept, consist of both aggressive (e.g., hurting others) and non-aggressive behaviors (e.g., stealing). It is still unclear whether air pollution would link to all kinds of immoral behaviors and we would examine their relationship in the current research. ”.
Question 2: While I believe it was correct to use regression, presenting the correlational matrix also helps. For example, it is of interest to know whether moral judgment and moral intention correlate or not.
Answer question: Thank you for your valuable suggestion. We have added the correlational matrix in the results part of both studies in our revised manuscript.
Question 3: Study 2. How were participants recruited? Air pollution varied a lot in China. Were participants recruited from difference regions? Additionally, did all participants complete the survey at the same time? If not, participants might experience difference levels of air pollution when they were completing the survey. These two factors might confound your findings. As an example, the variable of current weather was not significant in Study 1. However, the current weather, as indicated by AQI, was significant in Study 1. Thus, the findings appeared inconsistent between the two studies.
Answer question: Thank you for your valuable suggestion. Firstly, we have added detailed information about the recruitment of participants in our revised manuscript. The specific content is as follows: “Ninety participants (61 women; age 14–46, Mage = 26.97 years, SDage = 7.09) were recruited from a university in Beijing China for partial course credits. Participants were randomly assigned to either air pollution condition or control condition. They completed the tasks during the class and on computers. All the tasks were administered by Qualtrics.”
Secondly, the measures of current weather and AQI served for different purposes in Study 1. We’ve added the illustration of their distinctions in the main text to avoid any confusions. “It is noteworthy that the measure of weather is different from air pollution index because many factors other than air pollution (e.g., cloud, rain, and wind) can influence weather evaluation. We intended to measure and control for the general weather to observe the exclusive effect of air pollution. To support our surmise that the subjective evaluation of weather and air pollution is different, Table 1 showed that they only had a small and weak correlation (r = .10)”.
Question 4: Even you assigned someone to the control condition, what if someone recalled a day with air pollution?
Answer question: Thank you for your valuable question. We have carefully checked the participants’ answers in the manipulation and found no one in the control condition recalling a day with air pollution and no one in the experiment condition recalling a day without air pollution. In addition, we had a manipulation check and its results showed that the manipulation successfully made participants feel the weather as dirty. We’ve highlighted the manipulation effectiveness in the revised manuscript by adding a sentence noting that, “We have checked participants’ answers and found that all the participants had followed the instruction.”
Question 5: Abstract. You stated that “Study 1 used the object Air Quality Index to indicate the level of air pollution.” While your statement was correct, you should also indicate the main findings of study 1.
Answer question: Thank you for your valuable advice. We agree with you and have revised the abstract to the following: “The present research recruited participants from China, which is suffering from serious air pollution, and examined whether air pollution would be associated with moral judgment and immoral behavioral intention. Study 1 (n = 145) used the objective Air Quality Index to indicate the level of air pollution and found that it predicted harsh judgment on others’ moral violations but did not predict judgment on others’ non-moral negative behaviors or their own immoral behavioral intentions. Study 2 (n = 90) asked participants either to recall a past experience of being exposed to air pollution or to recall a neutral experience and consistently found that air pollution only influenced judgment on moral violations. The findings also ruled out the feeling of threat or the trust of government as possible mediators in the relationship between air pollution and harsh moral judgment. ”

Reviewer 2 Report
This short manuscript reports on the results of two studies in which the authors examine the predictive association of air quality (both actual and perceived) to moral judgment-making. They find that air quality is a significant predictor of harsher moral judgements.
While I think that the topic is interesting, this manuscript and the study it reports suffers from a number of problems and inadequacies which need to be addressed in a major revision before it can be given serious consideration for publication. My more substantive comments about the manuscript are below.
(1) Given the topic and research cited in the introduction, the authors should not only describe what was found in past studies but also report quantitative information regarding level of significance and effect sizes found in the earlier investigations. This will provide readers with more information to help them evaluate the extent to which any results are likely to be robust, replicable, and generalizable to the real world. It will also equip the authors with more information to help account for their own find of small effect sizes.
(2) There are no formal hypotheses given. Research expectations should be explicitly stated. From what I can ascertain, the authors are premising their study on the general belief that perceived dirtiness associated with air quality will itself be associated with greater negative affect (e.g., disgust) which, in turn, will influence moral judgement in such a manner that harsher and less tolerant judgements will be made. The authors should provide hypotheses for both studies.
(3) Power analysis should be computed and reported for both studies.
(4) On page 3, the authors report an alpha reliability coefficient for their single "threat" item. Cronbach's alpha cannot be computed for single items.
(5) On the page 3, the authors also report an alpha for the responses to the five moral vignettes. The alpha value is poor (.64). Given the nature of the vignettes, the use of Cronbach's alpha may be seen as less appropriate than a measure of inter-rater agreement.
(6) The rating of social class as reported on page 3 is an ordinal and not interval/ratio measure. As such, the median and not mean should be reported as the measure of central tendency.
(7) The authors need to provide zero-order bivariate correlations between all predictors and criterion variables and report the full regression results (i.e., unstandardized and standardized regression weights; constant/intercept; effect sizes for all significant predictors). They should also report confidence intervals around all regression weights and effect sizes (e.g., R-squared). Similarly, tables should be used to provide full descriptive statistical information about all variables used in study two and a full reporting of statistical results given.
(8) It is worth noting that the findings of study one and two are not completely consistent with past research. In particular, the authors note for both studies that negative mood was a significant predictor of greater acceptance of morally problematic behavior. Negative mood and emotionality (e.g,. anger, disgust) have been found to be associated with more critical/harsh judgements of others.
Housekeeping items-- Some of the writing could be improved to enhance clarity of expression. For example, on page 4 the following text appears--" Participants then reported their current mood. Except for the six items used in Study 1, two additional items were added (i.e., angry and anxious; 1 = not at all, 7 = very much). Scores on three positive-mood items (M = 4.05, SD = 1.47, α = .89) and five negative-mood items (M = 2.93, SD = 1.56, α = .93) were averaged respectively." It is not clear what is meant by "except for the six items used in study 1." From what I can determine, the authors should have stated "In addition to the six items used in study 1."
Thank you for allowing me to review this manuscript.
Author Response
Reviewer 2:Comments and Suggestions for Authors
This short manuscript reports on the results of two studies in which the authors examine the predictive association of air quality (both actual and perceived) to moral judgment-making. They find that air quality is a significant predictor of harsher moral judgments. While I think that the topic is interesting, this manuscript and the study it reports suffers from a number of problems and inadequacies which need to be addressed in a major revision before it can be given serious consideration for publication. My more substantive comments about the manuscript are below.
Question 1: Given the topic and research cited in the introduction, the authors should not only describe what was found in past studies but also report quantitative information regarding level of significance and effect sizes found in the earlier investigations. This will provide readers with more information to help them evaluate the extent to which any results are likely to be robust, replicable, and generalizable to the real world. It will also equip the authors with more information to help account for their own find of small effect sizes.
Answer question: Thank you for your valuable advice. According to your advice, we check the effect size of the previous studies, we added the demonstration in the discussion part. The specific content is: “. Comparing to the previous study’s effect size [6] (effect size ranging from 0.11 to 0.05), our findings had small effects (Study 1, f2 = 0.03; Study 2, f2 = 0.06). We can explain that Chinese citizens apply strategies to alleviate the influence of air pollution, and gradually get used to or even be indifferent of it.”
Question 2: There are no formal hypotheses given. Research expectations should be explicitly stated. From what I can ascertain, the authors are premising their study on the general belief that perceived dirtiness associated with air quality will itself be associated with greater negative affect (e.g., disgust) which, in turn, will influence moral judgments in such a manner that harsher and less tolerant judgments will be made. The authors should provide hypotheses for both studies.
Answer question: Thank you for your valuable advice. We agree with you and have added the hypothesis formally in the revised manuscript. The specific content is as follows: “State the hypothesis formally, building on the revealed association between physical dirtiness and harsh moral judgment and the finding that air pollution increases physical dirtiness (Zhang et al., 2012), we hypothesized that the severe air pollution would predict harsh moral judgment.”
Question 3: Power analysis should be computed and reported for both studies.
Answer question: Thank you for your valuable advice. We have followed your advice and added the power analysis to justify our sample size in the revised manuscript. The specific content is as follow, “We predetermined the required sample size for two studies by G*Power 3.1. Because there were no prior studies examining the association between air pollution and moral judgment, we estimated the effect size of Study 1 to be small (η2 = .05). Accompanying an α of .05 (two tailed) and a power of 0.80, the required sample size of Study 1 was 159. In Study 2, due to the manipulation of air pollution, we predicted its effect size to be increased (η2 = .10). With an α of .05 (two tailed) and a power of 0.80, the required sample size of Study 2 was 81. In addition, the data collection was further contingent on participants’ availability. We finally recruited 145 participants for Study 1 and 90 participants for Study 2.”
Question 4: On page 3, the authors report an alpha reliability coefficient for their single "threat" item. Cronbach's alpha cannot be computed for single items.
Answer question: Thank you for your comments. We used three items to calculate an index for the feeling of threat in the Study 1. The specific content is as follows, “Participants also evaluated how threatened, uncontrollable and unsafe (1 = not at all, 7 = very much) they felt at that moment. The scores were averaged to index the feeling of threat (M = 2.01, SD = 1.25, α = .90).” Thus it should be fine to report the alpha reliability coefficient for the feeling of threat.
Question 5: On the page 3, the authors also report an alpha for the responses to the five moral vignettes. The alpha value is poor (.64). Given the nature of the vignettes, the use of Cronbach's alpha may be seen as less appropriate than a measure of inter-rater agreement.
Answer question: Thank you for your suggestion. In the previous research where the moral vignettes adopted (Lammers, Stapel, & Galinsky, 2010), each moral vignette was used in different studies, thus the researcher did not report Cronbach's alpha, thus we cannot compare our alpha with existing studies. We also checked the literature about the definition of inter-rater reliability and found that it may not apply to the current research. Inter-rater reliability refers to the degree of agreement among different raters. It is a score of how much homogeneity, or consensus, there is in the ratings given by various judges and it is an indicator of validity. Our study asked the same participant to make judgment on different moral violations. If we understand correctly, the nature of the data don’t allow us to calculate the inter-rater agreement.
The poor alpha of these moral vignettes may be because each moral vignette represents a different aspect of morality. There are a lot of research and theories in moral psychology demonstrating that morality has different foundations (the moral foundations theory; Graham et al., 2013). The vignettes of “breaking traffic rules for a late appointment” may violate the authority/subversion foundation, while the vignette of “keeping a stolen and abandoned bike” may violate the fairness/cheating foundation.
Internal consistency (indicated by Cronbach's alpha in this research) measures whether several items that propose to test the same general construct based on the correlations between different items on the same test. It is an indicator of reliability. Because the moral vignettes used in the current research are linked to different moral domains, it is very possible that their Cronbach's alpha is not high. Maybe because of the same reason, some past research using different moral vignettes in the same study did not report Cronbach's alpha of these vignetts (e.g., the Study 2 of Polman & Ruttan, 2012). Similarly, other research using scenarios that measuring different things reported very low Cronbach's alpha (e.g., α = .06 in Study 5A of Zhou, He, Yang, Lao, & Baumeister, 2012).
Therefore, after meticulous consideration, we decide to keep the report of Cronbach's alpha and also explained the possible reason of Cronbach's alph in the manuscript, “The low Cronbach’s alpha may be because these moral vignittes gauged different aspects of morality.” We would be happy to make revisions in the next round if the reviewer has different suggestions.
Question 6: The rating of social class as reported on page 3 is an ordinal and not interval/ratio measure. As such, the median and not mean should be reported as the measure of central tendency.
Answer question: Thank you for your valuable advice. Past research has considered the rating of social class as a continuous variable and only reported mean as the measure of central tendency (e.g., Adler et al., 2000; Piff et al., 2012). We followed your suggestion and have reported both median and mean to in the revised manuscript.
Question 7: The authors need to provide zero-order bivariate correlations between all predictors and criterion variables and report the full regression results (i.e., unstandardized and standardized regression weights; constant/intercept; effect sizes for all significant predictors). They should also report confidence intervals around all regression weights and effect sizes (e.g., R-squared). Similarly, tables should be used to provide full descriptive statistical information about all variables used in study two and a full reporting of statistical results given.
Answer question: Thank you for your valuable advice. We followed your advice and have provided zero-order bivariate correlations between all predictors and criterion variables and reported the full regression results for both studies in this revised manuscript.
Question 8: It is worth noting that the findings of study one and two are not completely consistent with past research. In particular, the authors note for both studies that negative mood was a significant predictor of greater acceptance of morally problematic behavior. Negative mood and emotionality (e.g,. anger, disgust) have been found to be associated with more critical/harsh judgements of others.
Answer question: Thank you for your valuable advice. We have followed your advice and carefully examined the past literate related to negative emotion and moral judgment. First, not all negative emotions are associated with harsher judgments of others. For instance, Schnall et al. (2008) found that induced sadness (one item of the negative mood measure in the present research) could result in less harsh moral judgment when compared to the neutral condition. In other words, the relationship between negative emotion and moral judgment might vary as a function of each discrete emotion and moral domain (Horberg, Oveis, & Keltner, 2011; Ugazio, Lamm, & Singer, 2012).
Second, the past research on the relationship between negative emotion and moral judgment is about specific negative emotions, such as disgust and anger. However, the present research assessed a variety of negative emotions (e.g., sadness, dejection, and unhappy) and computed a general index for negative mood. Therefore, we did not hypothesize on the relationship between general negative mood and moral judgment and only controlled for it during the full regression analysis. We have added a detailed discussion on this issue in the revised manuscript. The specific content is as follows, “Lastly, both Studies 1 and 2 found that negative mood positively predicted the harsher judgment of others’ immoral behaviors, which seems inconsistent with past research. Previuos research showed that some negative emotions (e.g., disgust and anger) were associated with more critical or harsher moral judgment. However, not all negative emotions are associated with harsher judgments of others. For instance, Schnall et al. (2008) found that induced sadness (also measured in the present research) could result in less harsh moral judgment when compared to the neutral condition. In other words, the relationship between negative emotion and moral judgment might vary as a function of each discrete emotion and moral domain (Horberg, Oveis, & Keltner, 2011; Ugazio, Lamm, & Singer, 2012).
Besides, the past research on the relationship between negative emotion and moral judgment mostly focused on one specific emotion, such as anger and disgust. Few research, to the scope of our knowledge, has examined the impact of general negative mood on moral judgment. It might be an interesting direction for future research to test the relationship between general negative mood and moral judgment and the psychological mechanism.”.
Question 9: Housekeeping items-- Some of the writing could be improved to enhance clarity of expression. For example, on page 4 the following text appears—" Participants then reported their current mood. Except for the six items used in Study 1, two additional items were added (i.e., angry and anxious; 1 = not at all, 7 = very much). Scores on three positive-mood items (M = 4.05, SD = 1.47, α = .89) and five negative-mood items (M = 2.93, SD = 1.56, α = .93) were averaged respectively." It is not clear what is meant by "except for the six items used in study 1." From what I can determine, the authors should have stated "In addition to the six items used in study 1."
Answer question: Thank you for your valuable advice. We have carefully checked and improved the writing throughout the manuscript. As with the specific sentence you mentioned on page 4, we have revised it into “Participants then reported their current mood. In addition to the six items used in Study 1, two additional items were added (i.e., angry and anxious; 1 = not at all, 7 = very much). Scores on three positive-mood items (M = 4.05, SD = 1.47, α = .89) and five negative-mood items (M = 2.93, SD = 1.56, α = .93) were averaged respectively.”
Round 2
Reviewer 1 Report
Thanks for the authors' revision. All my previous comments have been addressed.
Author Response
Thank you for your valuable suggestions.
Reviewer 2 Report
Thank you to the authors for their responses to my comments on their original submissions. In general, I find the authors responses to be satisfactory. With that stated, I do have a couple of new comments on the revised paper.
(1) I appreciate the inclusion of a formal hypothesis. The new text on the bottom of page 1/top of page two giving the hypothesis states the following "State the hypothesis formally, building on the revealed association between physical dirtiness and harsh moral judgment and the finding that air pollution increases physical dirtiness[7], we hypothesized that the severe air pollution would predict harsh moral judgment." The words "State the hypothesis formally" can be deleted so that the sentence begins with "Building on the revealed association..."
(2) The first regression reported in study 1 and which uses acceptance of moral violations as the dependent variable, the authors report that hourly AQI emerged as a significant predictor in a manner consistent with their hypothesis. Examination of the confidence interval for the beta weight as reported in Table 2, however, reveals that it contains zero. This indicates that the predictor is, in fact, not significant. The fact that the zero-order bivariate correlation between hourly AQI and acceptance of moral violations is not significant is consistent with the hourly AQI being interpreted as a non-significant predictor.
In the second regression with acceptance of non-moral negative behavior as the dependent variable, negative mood is reported as statistically significant but its confidence interval also appears to include zero.
If the upper values for the confidence intervals for hourly AQI in the first regression and acceptance of non-moral negative behavior in the second regression only appear to include zero because the authors are reporting values to two decimal places, then I suggest that they report the values to three decimal places. However, if the upper values are truly zero, then the authors will need to reinterpret the regression results. The p-value relates to the exact point estimate of the beta weight and not the extent to which the beta weight for the sample corresponds to the weight for the population. When a confidence interval for an estimate includes zero, it often means that the result will not generalize to a population and will not be successfully replicated in future research.
(3) In Table 3, correlation coefficients are not reported in a consistent manner. No zero should be placed in front of the decimal and all coefficients should be rounded to two decimal places.
(4) In Table 3, I noted that the correlation between positive mood and the manipulation is reported as significant but the correlation between acceptance of moral violations and the manipulation is reported as non-significant even though the correlation coefficient appears to be the same value (i.e., -.26). This may be due to rounding. The authors are encouraged to double check the accuracy of the coefficients and significance reporting.
(5) A minor point-- In the newly inserted text on page 11 in which the authors report the range of effect sizes found in other research, they report "0.11 to 0.05." It is more conventional to report lower to higher values, i.e., 0.05 to 0.11. The authors may also want to explicitly state that the effect sizes they have found appear to be consistent with those found in past research.
Author Response
Reviewer2-round2:Comments and Suggestions for Authors
Question 1: I appreciate the inclusion of a formal hypothesis. The new text on the bottom of page 1/top of page two giving the hypothesis states the following "State the hypothesis formally, building on the revealed association between physical dirtiness and harsh moral judgment and the finding that air pollution increases physical dirtiness[7], we hypothesized that the severe air pollution would predict harsh moral judgment." The words "State the hypothesis formally" can be deleted so that the sentence begins with "Building on the revealed association..."
Authors response:Thanks for your valuable suggestions. We completely agree with you and have deleted the words “State the hypothesis formally” in the revised manuscript. The specific content is as follows, “Building on the revealed association between physical dirtiness and harsh moral judgment and the finding that air pollution increases physical dirtiness [7], we hypothesized that the severe air pollution would predict harsh moral judgment.”
Question 2: The first regression reported in study 1 and which uses acceptance of moral violations as the dependent variable, the authors report that hourly AQI emerged as a significant predictor in a manner consistent with their hypothesis. Examination of the confidence interval for the beta weight as reported in Table 2, however, reveals that it contains zero. This indicates that the predictor is, in fact, not significant. The fact that the zero-order bivariate correlation between hourly AQI and acceptance of moral violations is not significant is consistent with the hourly AQI being interpreted as a non-significant predictor.
In the second regression with acceptance of non-moral negative behavior as the dependent variable, negative mood is reported as statistically significant but its confidence interval also appears to include zero.
If the upper values for the confidence intervals for hourly AQI in the first regression and acceptance of non-moral negative behavior in the second regression only appear to include zero because the authors are reporting values to two decimal places, then I suggest that they report the values to three decimal places. However, if the upper values are truly zero, then the authors will need to reinterpret the regression results. The p-value relates to the exact point estimate of the beta weight and not the extent to which the beta weight for the sample corresponds to the weight for the population. When a confidence interval for an estimate includes zero, it often means that the result will not generalize to a population and will not be successfully replicated in future research.
Authors response:We appreciate the reviewer’s helpful comments. We agree with the reviewer’s concerns that 1) the pearson correlation between hourly AQI and acceptance of moral violations was non-significant (r = -.11, p = .198); 2) The upper bound of the confidence interval for the beta weight of hourly AQI was 0.
As for the first concern, pearson correlation was non-significant but the prediction of hourly AQI in the regression was significant because we included several confounding variables in the regression that decrease the residual errors.
As for the second concern, the reviewer is right that the upper values for the confidence intervals for hourly AQI in the first regression and negative mood in the second regression only appear to include zero because we only reported values to two decimal places. Following the reviewer’s suggestion and the published articles’ practice, we reported four decimals for all the confidential intervals (please refer to the manuscript for details, which are highlighted in red). After the revision, the two confidential intervals pointed out by the reviewer excluded 0. Specifically, the confidence interval for hourly AQI in the first regression is [-0.0111, -0.0003], and that for negative mood in the second regression is [-0.5743, -0.0014].
Question 3: In Table 3, correlation coefficients are not reported in a consistent manner. No zero should be placed in front of the decimal and all coefficients should be rounded to two decimal places.
Authors response:Thank you for the suggestion. We tried to follow the manner as the reviewer suggests. However, there are two cases that we don’t know how to deal with, the correlation between gender and task allocation (r = -.003) and that between wash and acceptance of moral violations (r =-.001). These two correlations should be -.00 if following the manner, thus we finally decided to leave three decimals for them. If the reviewer prefers to report -.00, we will be happy to make revision accordingly in the next round.
Question 4: In Table 3, I noted that the correlation between positive mood and the manipulation is reported as significant but the correlation between acceptance of moral violations and the manipulation is reported as non-significant even though the correlation coefficient appears to be the same value (i.e., -.26). This may be due to rounding. The authors are encouraged to double check the accuracy of the coefficients and significance reporting.
Authors response:Thank you for pointing this error out. The correlation between acceptance of moral violations and the manipulation should be significant, which has been rectified. Following the reviewer’s comments, we double checked all the correlations and confirmed that there are no other errors.
Question 5: A minor point-- In the newly inserted text on page 11 in which the authors report the range of effect sizes found in other research, they report "0.11 to 0.05." It is more conventional to report lower to higher values, i.e., 0.05 to 0.11. The authors may also want to explicitly state that the effect sizes they have found appear to be consistent with those found in past research.
Authors response:Thanks for your valuable suggestions. We completely agree with you and have modified the statements noting the summary of previous studies’ effect sizes in the revised manuscript. The specific content is as follows, “Consistent with the previous studies’ effect sizes [6] (ranging from 0.05 to 0.11), our findings had similarly small effects (Study 1, f2 = 0.03; Study 2, f2 = 0.06). One possible account for this small effect might be that Chinese citizens apply strategies to alleviate the influence of air pollution, and gradually get used to or even be indifferent of it.”
Round 3
Reviewer 2 Report
Thank you to the authors for responding to my comments on the revised manuscript.
Overall, I am satisfied with the re-revised manuscript.
One point of comment-- the authors indicated that they did not know what to do with two correlations that they report to three decimal places. I recommend that these be rounded to two decimal places. The fact that this will make the correlations appear as .00 or -.00 is generally understood by readers to mean that rounding has occurred.
Author Response
Question:One point of comment-- the authors indicated that they did not know what to do with two correlations that they report to three decimal places. I recommend that these be rounded to two decimal places. The fact that this will make the correlations appear as .00 or -.00 is generally understood by readers to mean that rounding has occurred.
Authors response: Thank for your advice. Based on you suggestion, we have revised the report format, which has been highlited in red in the revised manuscript.